# Influence of Intensive and Super-Intensive Olive Grove Management on Soil Quality—Nutrients Content and Enzyme Activities

**DOI:** 10.3390/plants12152779

**Published:** 2023-07-26

**Authors:** Marino Pedro Reyes-Martín, Emilia Fernández-Ondoño, Irene Ortiz-Bernad, Maria Manuela Abreu

**Affiliations:** 1Department of Soil Science and Agricultural Chemistry, Faculty of Science, University of Granada, Av. de Fuentenueva s/n, 18071 Granada, Spain; efernand@ugr.es (E.F.-O.); irene_ortizbernad@ugr.es (I.O.-B.); 2LEAF—Linking Landscape, Environment, Agriculture and Food Research Center, Associate Laboratory TERRA, Instituto Superior de Agronomia (ISA), University of Lisbon, Tapada da Ajuda, 1349-017 Lisboa, Portugal; manuelaabreu@isa.ulisboa.pt

**Keywords:** carbon content, nitrogen content, β-glucosidase, dehydrogenase, phosphatase, urease, protease, cellulase

## Abstract

Agricultural soil quality is an issue that has been widely debated in the literature in recent decades. Three olive grove areas (one in Lisbon and the others in Santarém, Portugal) with different management techniques (intensive and super-intensive) were selected. Nutrient concentrations and enzyme activities of soils were determined, as well as the C and N of litter and pruning waste (mulch) to estimate the influence of management techniques on the quality of olive grove soils and to assess the extent to which they are affected by organic covers and different cultivation intensities. Organic C and total N concentrations in soils of the intensive olive grove in Lisbon were the highest when compared with those in the intensive and super-intensive olive groves soils of Santarém. The concentrations of Ca, Mg, Na, and K were the main differences between the Lisbon olive groves and the other two from Santarém. Phosphatase, cellulase, and urease activities were related to the Na, extractable K, extractable P, Zn, Mn, organic C, and total N soil concentrations. Soil management and agricultural practices are determining factors for these enzymatic activities of Santarém olive groves, although climate conditions and soil properties play an important role in the soil enzymatic activities.

## 1. Introduction

The rural Mediterranean landscape is dominated by the olive crop (*Olea europea* L.). Worldwide, more than 10.9 Mha are grown, and the Mediterranean area accumulates 95% of this crop [1]. Specifically in Portugal, with almost 350,000 ha, it is one of the most extended crops on the land (CPP, Statistics of International Business 2014, Spanish Statistical Office).

Traditional crops (dry farming containing approximately between 50 and 160 trees per hectare) have been gradually replaced by intensive crops (400 trees per hectare and drip irrigation), and more recently by super-intensive crops (1200 and 2500 trees per hectare). Frequently, these olive groves, apart from being irrigated, are fertilized and structured to be pruned and harvested in a mechanical manner, thus increasing their productivity [1]. These techniques started to be implemented on a larger scale in the Iberian Peninsula during the mid-1990s [2], so their effects on the soil are still not accurately known.

According to Thoumazeau et al. [3], one of the main factors causing soil degradation is the crops, but the olive grove is the crop closest to naturalness in the Mediterranean environment [4]. However, the appropriate management of them is also one of the main factors that mitigate their degradation. The management of the olive crop has the capacity/potential to increase the soil content of organic matter [5], thus protecting the soil and contributing to retaining C [1]. Various activities for soil conservation have demonstrated the potential of olive groves as a stable organic C sink [1,6], and this management also affects the N dynamics [7]. However, it is still not accurately known to what extent the climate and the management affect the C production by the diverse types of olive groves [1], especially with new super-intensive crops. Activities performed in agricultural ecosystems may affect the soil quality through the crop techniques that are applied, among those the intensive management, which can negatively affect, modify, and possibly deplete soil functions [8,9].

Among the responsible agricultural methods, one can include the use of a cover crop [10]. One of the main pillars of conservative agriculture is having a permanent cover on the soil made with vegetation cover and/or pruning wastes. This cover influences the physical, chemical, and biological traits of the soil, thus fostering changes in the soil, which vary depending on its status [11].

Numerous studies focusing on cover crops in olive groves show their beneficial impact on the environment, including the reduction of nutrient losses and the enhancement of soil fertility by increasing water retention, enriching nutritional properties, and improving soil physical properties [12]. Blevins [13] showed that introducing a cover crop protects the soil from erosion whilst increasing its quality, as it increases its fertility and capacity for water retention. The studies by Gómez-Muñoz et al. [14] showed a long-term increase in C content in the soil. Accordingly, Thoumazeau et al. [3] pointed out that the chemical composition of the organic covers that are used also affects the physicochemical properties of the soil. In addition, the production and the nutritional status of the tree will be benefitted, thus increasing the content of organic matter and total N and P of the soil according to Boselli et al. [15]. Studies by Fernández-Hernández et al. [16] observed that neither the efficiency nor the quality of the olive oil was negatively affected by the application of mulch, and more recently, Tejada and Benítez [17] pointed out that the application of this organic matter to the soil increases the macro and micronutrient levels of olive tree leaves. However, other authors have warned of the need for the proper management of cover crops to avoid water competition with olive trees [18].

To estimate the changes that the management can implement on the quality of agricultural soils, the microbial activity can be analyzed through the enzymatic activity [19], as this activity plays a crucial role in several biological activities of the soil, as well as presenting a quick response to the changes that may occur in soil management [20,21,22,23,24]. Consequently, the evaluation of soil enzymatic activities is suitable as an early indicator of changes in the biological state of the soil. Several studies show the response of the enzymatic activities as a result of changes in management practices (such as plowing, fertilization, type of cultivation, etc.), as well as the influence of environmental conditions on soil functions [25,26,27]. Thus, different types of cover crops produce different responses in soil enzymatic activities, as well as in the soil microbial biomass and soil microbial community structure [28]. The results of Samuel et al. [29] showed that the addition of mineral fertilizers to organic fertilizers (green manure and farmyard manure) resulted in a significant increase in each soil enzymatic activity due to heightened plant biomass production, which, in turn, stimulated soil biological activity upon incorporation. These soil enzyme activities are fundamental regulators of litter decomposition and may significantly influence the fractions of labile organic C on the soil [30], the biological N fixation, and the immobilization of nutrients [31]. Hence, the study of the soil enzymatic activities can help to understand how the chemical composition of organic substrates can influence the microbial activity of the soil and, therefore, the mineralization of those substrates [17].

Among the most studied enzymes, β-glucosidase, urease, and phosphatase can be highlighted, which intervene in the C, N, and P cycles [32]. The organic C content of the soil is deeply correlated with the enzymatic activity β-glucosidase [33], as well as having been the most applied enzyme to evaluate to what extent the quality of the soil is affected by different types of crops [32]. The urease activity, which catalyzes the conversion of urea into ammonium and carbon dioxide, being involved in the mineralization of organic N [34,35], is commonly applied in studies on the fertility, quality, and impact of contaminants in the soils [32]. Nevertheless, the microbial biomass of the soil seems not to be significantly correlated with this enzyme. It is considered a very stable enzyme that is rarely influenced by meteorological factors such as temperature, humidity, or radiation [33], although it can be inhibited by fungicide contamination [36]. The phosphomonoesterase activity in the soil is influenced by soil depth [37], the application of wastes, and seasonal changes [38], as well as being involved in the phosphorus cycle [32]. The protease activity, which releases amino acids from proteins, plays an important role in the N cycle [39]. This enzyme is more present in cultivated soils than in soils without vegetation, and in soils with the application of vegetable wastes, as well as in unplowed soils [38]. The protease activity is lower in soils with lesser salinity and more CaCO_3_ content, as shown by Emran et al. [40]. Dehydrogenase activity has been widely used to assess the changes during the first stages of organic matter oxidation [23,33] and to evaluate the application of fresh wastes onto the soil [38].

Cellulase is significantly correlated with the organic C dissolved in the soil [23]. This enzyme increases its activity in soils with cover crop applications such as hay, as shown by Zhang et al. [30]. Additionally, these authors showed that the activity of the cellulose-degrading enzymes increased significantly with N fertilization.

There is still not much information on how different soil covers on olive groves and intensive and super-intensive management produce changes in the soil fertility and the evolution of its biodiversity. Hence, the aim of this study is to point out to what extent the different organic covers and the different cropping intensity rates affect the soil quality through the study of the enzymatic activities of the olive grove soil (dehydrogenase, β-glucosidase, urease, phosphatase, protease, and cellulase), organic C, and micro- and macro-nutrients.

## 2. Results

### 2.1. Soil Analysis

The soils of the U-intensive olive grove located on the campus of the Higher Institute of Agronomy in Lisbon collected under the olive trees (A) and in the adjacent line (L) showed the same pH value, which was remarkably lower than the pH determined in the soil samples collected in Santarém, both on intensive (I) and super-intensive (S) management. The electrical conductivity of the soil under olive trees of the U olive grove showed the highest values of all locations (Figure 1).

The organic C and total N concentrations in the soils of U olive grove were the highest when compared with those in the soils of Santarém (S and I). However, the C values were only statistically significantly different between the soils collected in lines of I and U olive groves. Regarding total N, the differences were significant between all soils of U and I olive groves, and between the samples from U and S lines (L). The C/N ratio does not present differences except on the samples collected in the lines of I and U olive groves.

There was no difference in the NH_4_^+^ content between olive groves or between locations, and the NO_3_^−^ concentration was only different between I and S olive groves under trees (A).

The total concentration of macronutrients (Ca, Mg, Na, and K) was the main difference between the olive groves of U and the other two from Santarém. The Ca content on the U olive grove was significantly lower than on the I and S olive groves, and the Na and K content on the U olive grove was significantly higher than on the other two olive groves. Extractable P in the soils of the U olive grove was statistically significantly higher than in the soils of the I and S olive groves (Figure 2).

The Fe concentration in I and U was higher in the samples from the lines than in the samples collected under trees whereas the opposite was observed for Cu (Figure 3). The Fe, Mn, and Zn concentrations were, in general, much higher in U samples than in the soils of the other two olive groves.

### 2.2. Litter and Mulch Analysis

The mulch collected in I presented the highest pH value (7.2), whereas the lowest pH value (6.3) was observed in the U litter. The electrical conductivity of the U litter samples was significantly higher than the same samples of the olive groves located in Santarém (I and S) (Figure 4).

The organic C concentration of the U litter samples is significantly lower when compared with the same samples from the other two olive groves, but the total N concentration does not show significant differences in the three areas of study. Consequently, the C/N ratio of the U litter is significantly lower than in the other two areas. The total N concentration, as well as the C/N ratio in the litter samples, is significantly higher than in the mulch on the I and S olive groves (Figure 4).

### 2.3. Soil Enzyme Activities

The enzymatic activities of the soils collected in the three studied olive groves are shown in Figure 5. Some of these enzymatic activities presented wide variations among the different olive groves. In general, cellulase, phosphatase, glucosidase, and urease activities are higher in U soils. The dehydrogenase activity presents few differences, which are only statistically lower under trees than in lines on the U olive grove, and it is also lower under trees on S olive grove than in the U olive grove. The protease activity was significantly lower on the lines on the U olive grove than in the soils of the other olive groves.

The result of the conglomerate analysis performed on the data of soil, litter, and mulch samples of the different olive groves indicates a clear differentiation between U and the other olive groves (S and I) (Figure 6) and a higher similitude between I and S olive groves.

As shown in Figure 7, the three olive groves are clearly differentiated from each other as regards the components of the litter and mulch samples, the soil properties, and the enzymatic activities. The distribution of the U samples is grouped together, influenced primarily by the orientation of the variable organic C, total N, and extractable K from the soil. The samples of the S area are generally located on the opposite axis to the U area samples. The Ca concentration and the protease enzymatic activity seem to condition the distribution of S olive grove samples. The I olive grove samples are located in the central part of the graph, as there are no great differences in the distribution of the line and under-tree samples.

The tendency of the enzymatic activities that are located on different axes is remarkable, with β-glusosidase, dehydrogenase, and cellulase on the one hand, and on the other hand, phosphatase, urease, and protease, as well as the relation of the dehydrogenase activity to the characteristics of the litter and mulch samples, and the cellulase and glucosidase with the N species (N-NO_3_^−^ and N-NH_4_^+^) in the soil. Protease activity is closely related to Ca, and urease and phosphatase activities are related to certain soil properties that differentiate the soils, such as their electrical conductivity or the Zn and Mg concentrations.

The pH of the vegetable samples (litter and mulch) is positively correlated with their organic C and N concentration and the C/N ratio. The scarce correlations of N are remarkable. The organic C in the vegetable samples is negatively correlated with the organic C and total N of the soil (Figure 8).

The pH of the soil samples is negatively correlated with the electrical conductivity of both samples of vegetable samples (litter and mulch) and the soil concentrations of extractable K and P, and Na. The C/N ratio indicates that the mineralization/humification capacity of the vegetable samples (litter and mulch) is negatively correlated with the organic C and total N concentrations in the soil. Organic C and total N concentrations are positively correlated with extractable P, K, Na, Mn, and Zn. The Fe, Zn, Na, organic C, N-NH_4_^+^, extractable P, extractable K, Mn, and Zn concentrations in the soil are positively correlated with urease activity.

In general, phosphatase, cellulase, and urease activities are positively correlated with each other, as well as with concentrations of Na, extractable K, extractable P, Zn, Mn, and organic C and N in the soil. However, these enzymatic activities are negatively correlated with protease. The glucosidase activity presents positive correlations with the Fe, Mn, Zn, and Na concentrations, and negative correlations with the soil pH and the Ca and Cu concentrations. Protease activity is positively correlated with the Ca concentration of the soil and negatively correlated with the concentrations of Mn, Zn, P, K, and N-NO_3_. The cellulase activity presents positive correlations with the concentrations of organic C, N, P, K, Na, Mn, Zn, and Fe of the soil, and negative correlations with the concentrations of Ca, Cu, Mg, and the soil pH.

## 3. Discussion

Despite comparing two intensive management with one super-intensive management, the evolution of the vegetative cover due to the climatic conditions, management time, and the use of farming machinery affects soil quality, as can be seen in Figure 7. The U olive grove has a higher mean annual minimum temperature and a higher mean annual accumulated precipitation than the I and S olive groves, which may determine that the cover crop of U is the most developed. This is a consequence of the warmer temperatures as colder temperatures can contribute to a reduction in cover crops development. In addition, cover crops are better adapted to warm areas with abundant precipitations [41]. The samples studied in each area are grouped together near each other (Figure 7), and the U area is clearly different compared to I and S, which could perhaps be due, among other factors, to the differences between the rock-forming soils such as basalt in soils from U and marly limestones in I and S soils.

According to Bahnmann et al. [42], the pH affects the availability of nutrients such as Na, K, Ca, and Mg, and the microbial communities in the soil. Moreover, the studies carried out by Niu et al. [43] showed that the soil pH was negatively correlated with the available fraction of Fe, Mn, and Cu in the first 20 cm of soil. These findings partially match the results obtained, where the soil pH is negatively correlated with the analyzed micronutrients (Fe, Mn, and Zn) but not Cu. The Cu content was higher under trees than on the lines, likely due to having applied copper sulphate as a plant-protection product (Table 1), as was also pointed out by Samarajeewa et al. [44].

The organic C concentration of the soils was positively correlated with the Na and K concentrations, with the results being similar to those obtained by Lemanowicz and Bartkowiak [48]. In general, the C and N values in the soils of all three olive groves match those of González-Rosado et al. [49] for no-tillage olive groves. In the U area, the highest concentrations of C and N in the soils were observed, but the U area only had significant differences compared to the I area in C in lines and N in lines and under trees. The cover crop generates high N content, which can be easily decomposed, being available for absorption by the crop [28]. Some experiments conducted on intensively cultivated and irrigated olive groves have shown a substantial positive correlation between N fertilization and tree performance [12]. The cover crop in U is continuous (present even under trees), and the vegetation is taller and denser. The extension and conservational traits of the vegetative cover improve the soil quality, as was also observed by Niu et al. [43]. In addition, in U, the cover crop is more than six years old and showed the highest C and N concentrations, which matches the studies of Feng et al. [28], where soil health improved due to C, N, P, and S cycling when a cover crop was used for at least six years.

The enzymatic activities of the studied soils (Figure 4) were similar to those observed by other authors [32,33], and they varied widely in accordance with the soil type and the management practices (Figure 5). The phosphatase activity was positively correlated with the C content, which matches the studies by Paz-Ferreiro et al. [50]. Furthermore, it was correlated with N and P concentrations, as pointed out by Baldrian et al. [51] and Micuti et al. [23], respectively, K, and especially with Mn, Zn, and Na in the soil. This activity also showed negative correlations with the Ca and Mg concentration, particularly with the soil pH and the vegetable sample (litter and mulch) pH, which matches the studies by Baldrian et al. [51], pointing out that this enzyme had a negative correlation with the soil pH. This explains why the phosphatase activity is higher in U, where pH is more acidic and the content of available P is higher, which is in accordance with Hesterberg’s [52] results. In fact, the concentrations of extractable K and P in the U olive grove are also markedly higher than in the other areas of study.

In U, no pruning wastes were applied, but it has a continuous and well-developed cover crop and a C/N ratio that is significantly lower than in the other locations, so this litter can be expected to decompose more quickly [53] and the biological activity would be higher; in fact, it presents the highest activities of glucosidase, urease, and phosphatase because a more developed cover crop helps in improving these soil enzymatic activities [28]. The higher phosphatase activity in U may be due to the higher phosphorus and organic matter contents in the soil, compared with the other locations. According to various authors [21,54,55], the humidity complexes of the soil retain phosphatases, so these retained enzymes are more resistant to thermal denaturalization and protein degradation than free enzymes. Separately, when the organic matter of the soil increases and there is the presence of a vegetative cover, especially if it is a succession of natural vegetation, the urease activity increases [33] as occurs in the soils from U. As stated by Tejada and Benítez [17], the application of organic matter to the soil significantly stimulated its enzymatic activities.

Dehydrogenase activity has little correlation with the rest of the enzymatic activities and the edaphic and vegetable samples (litter and mulch) traits that have been studied (Figure 7). This is possibly due to the presence of Cu, which interferes with the measure of this activity (by reducing the TPF or INTF absorbance), so it is not trustworthy for Cu-contaminated (phytosanitary products) soils [33].

According to Brzezińska et al. [39], the protease activity is positively correlated with the C and N content. However, in U olive grove soils, despite having a higher content in C and N and a higher vegetative cover, the protease activity presents the lowest values, and this happens because the soils from the U olive grove present the highest Na content as well, which matches the studies of Emran et al. [40], which showed that this activity is lower on soils with a higher Na content.

The urease activity, as described by García et al. [33], presented a positive correlation with the N and organic matter concentrations in the soil. In addition, the glucosidase activity showed a positive correlation with the organic carbon content, which also matches the studies by García et al. [33].

Our results provide the importance of management in soil health, although a longer-term study with a greater number of sampled olive groves is necessary to verify the effect on production.

## 4. Materials and Methods

### 4.1. Study Area Location and Sampling

To evaluate the effects of the management on the soil quality, three olive grove areas in Portugal were selected. The first olive grove was located in the Santarém region (I), approximately 85 km northeast of Lisbon. The second olive grove was located on the campus of the Higher Institute of Agronomy of the University of Lisbon (U). Both olive groves had intensive management and various types of vegetation cover. The third one was also located in the Santarém region, and it had super-intensive management (S). All crops had litter (4 cm thick), an almost flat surface, a cover crop (trimmed every other year), there was no tillage, and the distribution of rainfall over the year was concentrated between autumn and spring.

The cover crop of the U olive grove was all over the surface of the crop, and it was substantially higher in volume and density compared with the I and S olive groves. On I and S olive groves, there was no cover crop under trees (only on the lines), and it was composed primarily of briofits (Figure 9). In addition, on I and S olive groves, pruning waste was applied to the lines every other year. The other traits and the description of crop management, as well as the treatments performed during the weeks prior to the collection of samples, can be found in Table 1.

### 4.2. Sampling Design

In February 2020, a homogeneous sample zone was selected in each of the studied olive groves, which was representative of the whole area. Four trees from different rows were selected, and samples of the litter under each tree were collected, as well as of the pruning waste (mulch) applied on the lines next to the tree, which correspond to the line between the tree row (Figure 9). In the U crop, no pruning waste (mulch) samples were collected as this waste was not used in this olive grove.

Under the selected four trees (A) and in the lines next to them (L), soil samples of the first 10 cm depth were collected. Each of the eight soil samples per olive grove came from a sample composed of four subsamples collected under the olive tree and adjacent line. The samples were transported to the laboratory in a box cooler, and after being sieved (<2 mm) and homogenized, each one was subdivided into two subsamples, one for the physicochemical analyses and the other for the analyses of enzymatic activity. Each of the analyses was performed in duplicate.

### 4.3. Soil Samples Analysis

Soil pH was determined with a glass electrode using a soil:water ratio of 1:2.5 (Jackson [56], 1964), and the same ratio was used for the electrical conductivity determination. Total N was determined by a semi-micro Kjeldahl digestion procedure [57], and nitrate (N-NO_3_) and ammonium (N-NH_4_) N were determined by automatic digestion and distillation (Foss™ Tecator Digestor™ 2508 and Foss™ Kjeltec™ 8200 Auto Distillation Unit, FOSS, Hilleroed, Denmark). After extraction using the Egner–Riehm method, extractable K was determined by atomic absorption spectrophotometry and extractable P was determined by colorimetry by the Olsen method [58]. The determination of total organic C (TOC) was carried out by the Walkley and Black [59] method via oxidation with potassium dichromate. The micronutrients Fe, Cu, Mn, and Zn and the macronutrients Ca, Mg, and Na were extracted by the Lakanen and Erviö [60] method and determined by atomic absorption spectrophotometry. The macronutrients Ca, Mg, and Na were also determined in this same extraction solution.

### 4.4. Litter and Mulch Analysis

To determine organic C, total N, pH, and the electrical conductivity of the vegetable samples (litter and mulch), they were passed through a chipper-shredder and mixed intensively to obtain a homogeneous composition. The shredded samples had an approximate maximum particle size < 5 mm.

Each homogeneous sample was dried for 48 h in a forced-air oven at 105 °C. Subsequently, the electrical conductivity and pH were measured on the filtered solution from 10 g of vegetable sample in 50 mL deionized water after being shaken and standing for 24 h. The organic C and total N were determined using the same methodology as for the soil samples.

### 4.5. Analysis of Soil Enzymatic Activities

The samples for enzymatic activity analysis (dehydrogenase, β-glucosidase, urease, phosphatase, protease, and cellulase) were kept moist in the fridge (4 °C). Prior to analyses, the soil samples were homogenized and passed through a 2 mm sieve and then split into two subsamples: One for humidity determination and the other was kept moist for enzymatic activity analysis (in triplicate).

To determine the β-glucosidase and phosphatase activities, the Tabatabai [61] and the Tabatabai and Bremner [62] methods were applied, respectively. The method used for β-glucosidase activity is based on the colorimetric determination of *p*-nitrophenol generated from the β-glucosidase reaction. This reaction takes place after incubating the soil with the β-D-glucopyranoxide substrate in a buffered medium at pH 6. This process takes place during 1 h of incubation at 37 °C. The method used for the determination of phosphatase activity is based on the spectrophotometry of *p*-nitrophenol released when the soil is incubated at 37 °C for 1 h with a buffered *p*-nitrophenylphosphate solution. The results were expressed in µmol p-Nitrophenol g^−1^ dry soil matter h^−1^. The Tabatabai [63] method was applied to determine dehydrogenase activity. This method estimates the amount of iodonitotetrazoyl formazan formed in the soil after being incubated in the dark with 2-*p*-iodophenyl-3-*p*-nitrophenyl-5-phenyltetrazoyl for 20 h at 20 °C. The results were expressed in µg TPF g^−1^ dry soil matter 16 h^−1^. To determine urease activity, the Kandeler and Gerber [64] method was used. This method is based on the colorimetric determination of the ammonium released in the incubation of a soil solution at 37 °C for 2 h. The reaction of ammonium in an alkaline medium with a coloring agent provides monochloramine, which, in the presence of thymol, is transformed into a colored propyl compound whose color intensity is proportional to its concentration. The results were expressed in µmol N-NH_4_^+^ g^−1^ dry soil matter 2 h^−1^. The Hope and Burns [65] method was followed to determine cellulase activity. This method is based on the measurement of glucose release from a cellulose substrate. The soil sample was incubated with a microcrystalline cellulose substrate (Avicel), in a buffered solution at pH 5.5 and 40 °C for 16 h. The results were expressed in µmol glucose g^−1^ dry soil matter 16 h^−1^. The Ladd and Butler [66] method was used to determine protease activity. This method is based on the colorimetric determination via the Folin reaction of the released peptides soluble in trichloroacetic acid after incubating the soil with casein for 2 h at 50 °C. The results were expressed in µmol tyrosine g^−1^ dry soil matter 2 h^−1^.

### 4.6. Statistic Analysis

The analysis of the distribution of each variable was performed using the Saphiro test. For the determination of the differences between samples from different olive groves, the non-parametric Kruskal–Wallis test was performed. Bivariate correlations were performed according to the Pearson method. The information of multiple dimensions was visualized and interpreted with non-metric multidimensional scaling (NMDS). Principal Component Analysis (PCA) was performed using all studied variables. The significance level was set at a *p*-value < 0.05. All computations were made using the IBM SPSS Version 20.0 [67] and RStudio 2015 [68], using dplyr, rstatix, ggpubr, and corrplot as the main Bioconductor packages.

## 5. Conclusions

Different cultivation intensities and agricultural practices (cover crop, mulch, fertilizers, etc.) produce sensitive changes in the enzymatic activities of the soil. The urease, glucosidase, phosphatase, and protease enzymatic activities showed great differences among the olive groves with different management and cultivation intensities and differences in the soil quality (e.g., pH, total N concentration, and extractable K and P). The U olive grove (intensive) presented the highest concentration of C, N, P, K, Fe, Mn, and Zn and the highest cellulase, phosphatase, β-glucosidase, and urease enzymatic activities. Despite not having mulch application, it is the crop with the most visibly developed cover crop. Soil management and agricultural practices are determining factors for the enzymatic activities of intensive and super-intensive olive groves. Climatic conditions and the properties of the soils also play an important role in the soil quality increase as is the case of the U intensive olive grove. However, to be correctly interpreted, enzyme activities must be studied along with other soil properties.

## Figures and Tables

**Figure 1 plants-12-02779-f001:**
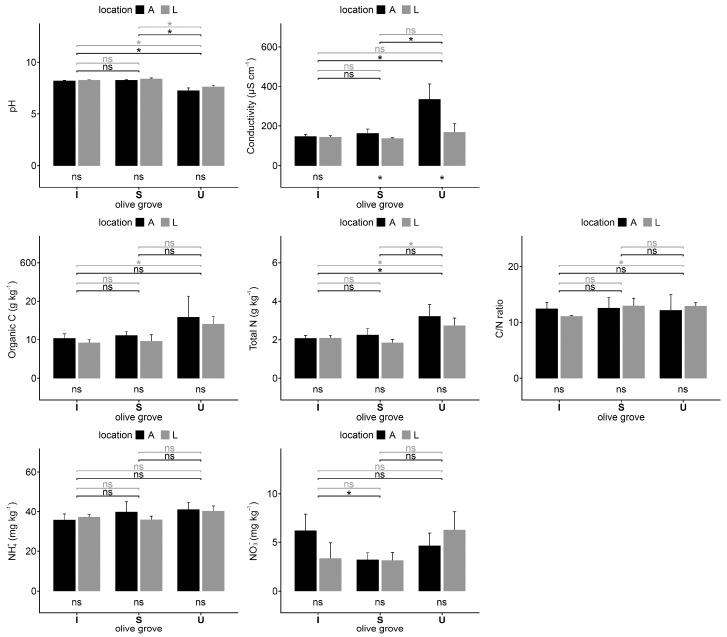
Electrical conductivity, pH, organic C, total N and C/N ratio, N-NO_3_^−^, and N-NH_4_^+^ of soil samples collected under trees (A) and in the lines (L) in each olive grove: Intensive (I), super-intensive (S), and intensive-university (U). Vertical thin bars represent the standard deviation of the sample (*n* = 4). The horizontal lines indicate differences between samples under tree (black) and between lines (grey) on different olive groves. The differences between A and L samples in the same olive grove are indicated under the bars. * = significant differences (*p* < 0.05) according to the Kruskal–Wallis test; ns = no significant differences.

**Figure 2 plants-12-02779-f002:**
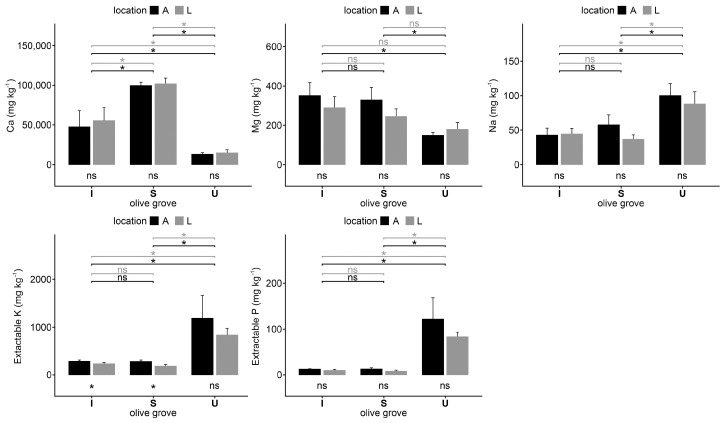
Concentrations of Ca, Mg, Na, and extractable K and P of soil samples collected under olive trees (A) and in the lines (L) in each olive grove, intensive (I), super-intensive (S), and intensive-university (U). Vertical thin bars represent the standard deviation of the sample (*n* = 4). The horizontal lines indicate differences between under tree (black) and between lines (grey) on different crops. Within the same crop, the differences between under tree and lines are indicated under the crop bars. * = significant differences (*p* < 0.05) according to the Kruskal–Wallis test; ns = no significant differences.

**Figure 3 plants-12-02779-f003:**
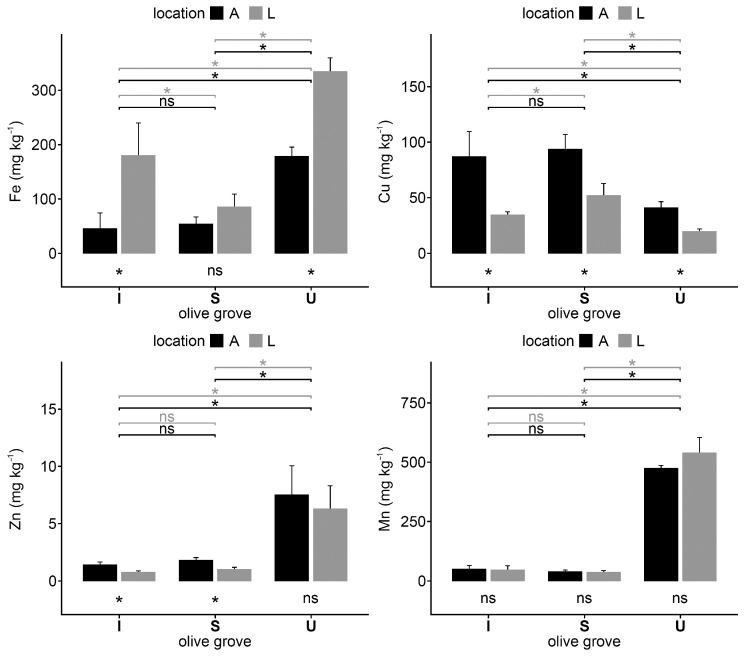
Concentrations of the micronutrients Fe, Cu, Mn, and Zn of soil samples collected under olive trees (A) and in the lines (L) in each olive grove; intensive (I), super-intensive (S), and intensive-university (U). Vertical thin bars represent the standard deviation of the sample (*n* = 4). Horizontal lines indicate differences between under tree (black) and lines (grey) among different crops. The differences between under tree and street within the same crops are indicated under the crop bars. * = significant differences (*p* < 0.05) according to the Kruskal–Wallis test; ns = no significant differences.

**Figure 4 plants-12-02779-f004:**
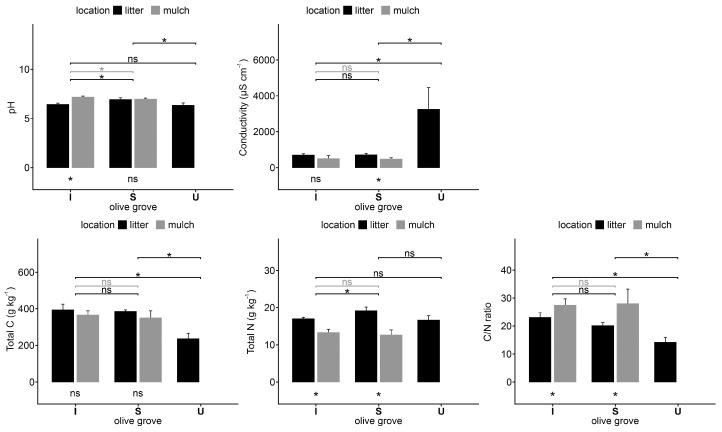
Electrical conductivity, pH, C, and N concentrations and C/N ratio of litter samples collected under olive trees (litter) and pruning waste (mulch) samples from the lines in each olive grove; intensive (I), super-intensive (S), and intensive-university (U). Vertical thin bars represent the standard deviation of the sample (*n* = 4). Horizontal lines indicate differences between litter (black) and mulch (grey) samples in different olive groves. Within the same area, the differences between under tree and line samples are indicated under the olive grove bar. * = significant differences (*p* < 0.05) according to the Kruskal–Wallis test; ns = no significant differences.

**Figure 5 plants-12-02779-f005:**
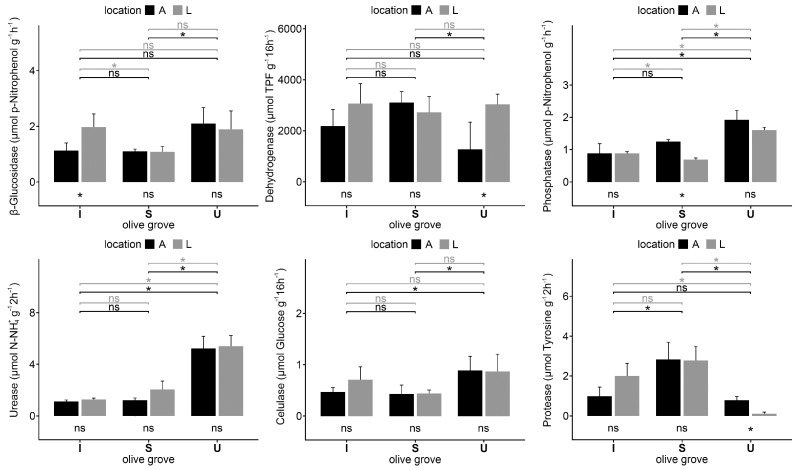
Enzyme activities of soil samples collected under olive tree (A) and on the lines (L) in each olive grove; intensive (I), super-intensive (S), and intensive-university (U). Vertical thin bars represent the standard deviation of the sample (*n* = 4). Horizontal lines indicate the differences between under tree (black) and lines (grey) of different olive groves. Within the same olive grove, the differences between samples collected under tree and line are indicated under the olive grove bars. * = significant differences (*p* < 0.05) according to the Kruskal–Wallis test; ns = no significant differences.

**Figure 6 plants-12-02779-f006:**
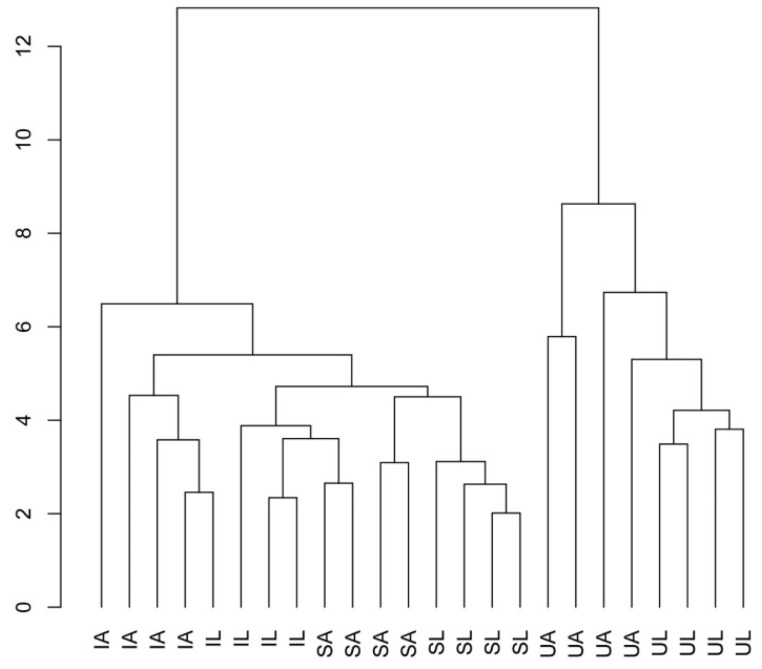
Dendrogram of the cluster analysis carried out with all the variables determined in the soil, litter, and mulch samples of the different olive groves (under tree on intensive crops in the Santarém region (IA), line in intensive management in the Santarém region (IL), under tree on super-intensive management in the Santarém region (SA), line on super-intensive management in the Santarém region (SL), under tree on intensive olive grove in the University of Lisbon (UA), and line on intensive management in the University of Lisbon (UL)).

**Figure 7 plants-12-02779-f007:**
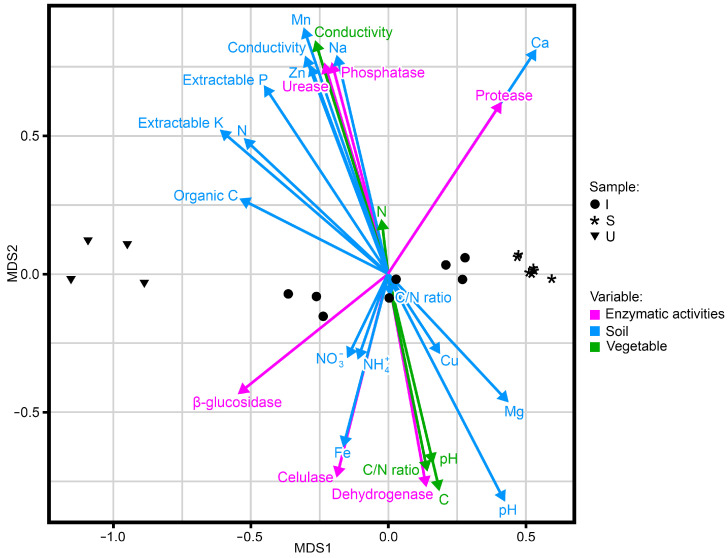
Non-metric multidimensional scaling (NMDS) plot of all the samples analyzed, including all the variables studied and their relation to the soil enzyme activities.

**Figure 8 plants-12-02779-f008:**
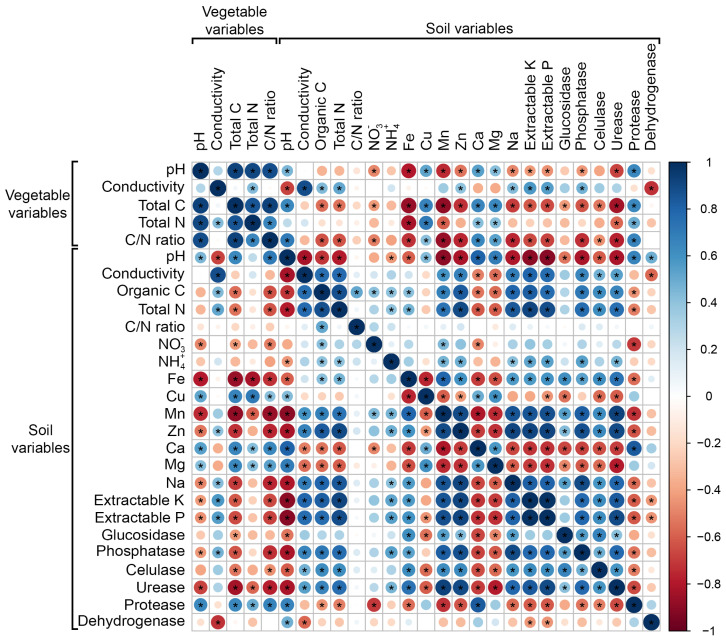
Clustering of multivariate soil and vegetable samples (litter and mulch) data using Principal Component Analysis and heatmap. * = significant differences (*p* < 0.05).

**Figure 9 plants-12-02779-f009:**
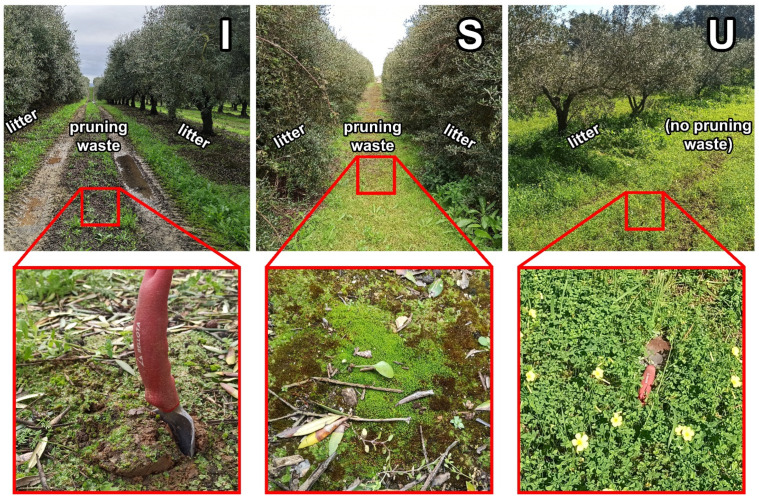
General overview of olive groves and detailed view of the soils of Santarém region with intensive (I) and super-intensive (S) management, and of the Higher Institute of Agronomy (University of Lisbon), with intensive management (U). Red squares show a close-up of the soil surface.

**Table 1 plants-12-02779-t001:** Traits and management practices of the three studied olive groves.

Olive Groves	Santarém Intensive (I)	Satarém Super-Intensive (S)	University Intensive (U)
Coordinates	39°16′40.1″ N 8°39′40.7″ W	39°17′36.4″ N 8°41′24.0″ W	38°42′52.8″ N 9°11′05.6″ W
Original material	marly limestones	marly limestones	Basalt
Height	35 m s.n.m.	56 m s.n.m.	123 m s.n.m.
Bioclimatic stage *	Mesomediterranean	mesomediterranean	thermomediterranean
Annual average minimum temperature **	11.22 °C	11.22 °C	13.57 °C
Annual average maximum temperature **	22.86 °C	22.86 °C	21.86 °C
Annual average accumulated precipitation **	609.33 mm	609.33 mm	815.23 mm
Soil classification ***	Calcium Cambisol	Calcium Cambisol	Chromic Vertisol
Area	22 ha	80 ha	0.9 ha
Age of the crop	≈35 years	20 years	35 years
Management (framework of the soil)	Intensive (7 × 5 m)	Super-intensive (1.35 × 3.75 m)	Intensive (7.5 × 5 m)
Variety	Picual	Arbequina	Picual
Application of pruning waste (mulch) on the lines	left on the soil	left on the soil	exported
Fertilizer	potassium nitrate, liquid potassium with EDTA	granulated ammonium sulfate	potassium sulfate with magnesium, ammonium sulfate, ammonium nitrate
Cover crop management	brush cutter	brush cutter	brush cutter
Foliar fertilizer	phosphorus pentoxide, potassium oxide, magnesium oxide	B, K nitrate, Mg nitrate, Fe	sodium borate, dimethoate, copper phosphite
Fungicide and pesticide	20% copper (copper calcium sulfate)	tebuconazole	difenoconazole, trifloxystrobin, copper and calcium sulfate
pH regulator	16% nitrogen and 41% sulfur trioxide	16% nitrogen and 41% sulfur trioxide	does not need
Adjuvants	metoxi poli(etoxi)-propil heptametiltrisiloxano	metoxi poli(etoxi)-propil heptametiltrisiloxano	does not need

* [45]. ** Recorded from 1999 to 2018 [46]. *** [47]. The quantities of fertilizer and foliar fertilizer applied to the three olive groves were similar.

## Data Availability

The data presented in this study are available upon request from the corresponding author.

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
