# Peer review of "Influence of Intensive and Super-Intensive Olive Grove Management on Soil Quality—Nutrients Content and Enzyme Activities"

_plants, 2023, doi:10.3390/plants12152779_

Round 1

Reviewer 1 Report

Olive grove management and soil quality

            In this study the authors aimed to estimate the changes that management techniques can cause on the quality of olive grove soils. They analyzed the enzymatic activities and nutrients of these soils to assess the extent to which they are affected by the organic covers and different cultivation intensities. The authors selected 3 olive grove areas with different management techniques. The first olive grove was located in the Santarem region (I), about 85 km northeast of Lisbon. The second olive grove was located in the campus of the Higher Institute of Agronomy of the University of Lisbon (U). Both olive groves had intensive management and various types of vegetation cover. The third one was also located in the Santarém region (S), and it had super-intensive management. The authors concluded that soil management and agricultural practices are a determining factor for the enzymatic activities of I and S olive groves although the climatic conditions and the properties of the soil play an important role in the enzymatic activities of the soil contributing to increase the soil quality as it happens with the U olive grove.

The MS is well written and the results are well presented and discussed. However, there are few comments and suggestions that may improve the manuscript:

-         Line 13-14. Three olive grove areas with different management techniques were selected. It would be better if the authors mentioned the names of the three areas as they reported them in Materials chapter. They already mentioned one of them (Santarém) in Line 17. So, for clarity I recommend authors to mention the rest of two areas.

-         Line 27. Olea Europea should be Olea europea and in etalic form. (Olea europea).

-         Line 88-89. Please merge the two paragraphs in one.

-         The same also in Line 98 -99 and 105-106.

-         In figure captions such as 153, 167, and elsewhere there is no need to cite any reference. The method with reference id already cited in Materials chapter.

-         Line 349. Is Santarem area different from Santarém in line 353? Or it should be Santarém.

-         In Table 1. Line 369. Traits and management practices of the three studied olive groves.  The table contained the type of fertilizers but without quantities. Are the three regions received equal quantities?

-         The same also is for foliar fertilizers. The type and level of fertilizers, if different, will affect the results.

-         Line 379. soil samples of the first 10 cm depth were collected. Why the samples were collected at 10 cm depth? I think from tree farms the soil samples are collected at 20 or 30 cm depth.

-         Line 398. The macronutrients Ca, Mg and Na were also determined in this same extraction solution. Please add some details about the methods used for Ca, Mg and Na   determination.

-         Line 402. (Vegetable samples). This expression is very confusing. What do you mean by vegetable samples? Normally, vegetable is vegetable crops. Or you mean vegetative. This expression is repeated several times throughout the MS such as line 244, 246, ……... Please revise and clarify.

-         Line 412. Analysis of soil enzymatic activities. This part need addition of some details about the methods used for determination of enzyme activities.

-         Line 349. Conclusions. This part need revision and please write it as one paragraph and focus only on your result and your message that you want to deliver.

Author Response

We would like to thank the reviewer for the comments to help us to improve the quality of the manuscript. Attached we send a Word document with all the answers.

Reviewer 2 Report

The authors have focused on the Olive grove management and soil quality. Interesting and well-developed Results part. Please see below my suggestions:

First sentence of the abstract is too general. Moreover, I am not agreeing with it – hundreds of papers are published in the field of soil management, for different crops, areas, climate, etc. Please reshape.

Latin names of species should be indicated in Italics. Please revise the entire manuscript, beginning with L27. Please check https://doi.org/10.1186/s43008-020-00048-6

L103. Chemical formulas should be written using subscript for coefficient (CaCO3) and using superscript for the charge of the ions (i.e. L143-144: NH4+; NO3-). Please proceed and revise the entire manuscript in this regard.

Table 1. Complete the first cell (head of the Table) it. They are not allowed empty cells in a table. Also, please unbold the first column (instructions for authors – Table setting). 

As the unit of measure for volume, ml should be replaced with mL, as Litter is the international unit of measure for volume. Revise the entire manuscript in this regard, correcting all the multiples or submultiples used to express the volumes.

Subsection 2.6. The statistic softs/computer programs used in this research must be mentioned, with their used variants, and referenced. Moreover, statistical parameters should be explained.

Discussion section:

-       How it can be improved the olives orchard management under the effects of climate change? I suggest checking and referring to https://doi.org/10.1007/s11356-019-04214-1 and https://doi.org/10.1007/s11356-021-14127-7

-       Regarding the effects of long-term application of organic and mineral fertilizers on soil enzymes, you should check https://doi.org/10.37358/RC.18.10.6590

-       Also, importance of nanotechnology should be provided in a paragraph, regarding the aspect of nano-farming versus nanotoxicity – please see https://doi.org/10.1016/j.chemosphere.2021.132533

-       After L345, before section 4, please add the strengths and the weakness of your research, in a LAST paragraph of Discussion.

-       To what extent the limitations of your research can be concretely resolved in future research directions?

References must be inserted in the MDPI style – check the Instructions for authors.

English is fine, few minor editing mistakes.

Author Response

(The authors gave the same response as above.)

Author Response

(The authors gave the same response as above.)

Round 2

Reviewer 2 Report

The authors responded to my requests.

Good English

Author Response

Dear Reviewer,

We would like to thank you for giving us a chance to revise the manuscript. Your feedback helps us to improve the quality of our work. Following your and the editor's comments, we have provided more specific responses to the reviewers' comments and changed the title to one more appropriate to the findings of the manuscript.

Yours faithfully,

The authors